# Mitochondrial-Targeted Therapies Require Mitophagy to Prevent Oxidative Stress Induced by SOD2 Inactivation in Hypertrophied Cardiomyocytes

**DOI:** 10.3390/antiox11040723

**Published:** 2022-04-06

**Authors:** Victoriane Peugnet, Maggy Chwastyniak, Paul Mulder, Steve Lancel, Laurent Bultot, Natacha Fourny, Edith Renguet, Heiko Bugger, Olivia Beseme, Anne Loyens, Wilfried Heyse, Vincent Richard, Philippe Amouyel, Luc Bertrand, Florence Pinet, Emilie Dubois-Deruy

**Affiliations:** 1Univ. Lille, Inserm, CHU Lille, Institut Pasteur de Lille, U1167-RID-AGE-Facteurs de Risque et Déterminants Moléculaires des Maladies Liées au Vieillissement, 59000 Lille, France; victoriane.peugnet@hautsdefrance.fr (V.P.); maggy.chwastyniak@pasteur-lille.fr (M.C.); steve.lancel@univ-lille.fr (S.L.); olivia.beseme@pasteur-lille.fr (O.B.); wilfried.heyse@inria.fr (W.H.); philippe.amouyel@pasteur-lille.fr (P.A.); 2Normandie Univ, UNIROUEN, Inserm U1096, FHU-REMOD-HF, 76000 Rouen, France; paul.mulder@univ-rouen.fr (P.M.); vincent.richard@univ-rouen.fr (V.R.); 3Pole of Cardiovascular Research, Institut de Recherche Expérimentale et Clinique, UCLouvain, 1200 Bruxelles, Belgium; laurent.bultot@uclouvain.be (L.B.); natacha.fourny@uclouvain.be (N.F.); edith.renguet@uclouvain.be (E.R.); luc.bertrand@uclouvain.be (L.B.); 4Department of Cardiology and Angiology I, Heart Center Freiburg, Faculty of Medicine, University of Freiburg, 79085 Freiburg, Germany; heiko.bugger@universitaets-herzzentrum.deres; 5Univ. Lille, CNRS, Inserm, CHU Lille, Institut de Recherche Contre le Cancer de Lille, UMR9020-UMR-S 1277-Canther-Cancer Heterogeneity, Plasticity and Resistance to Therapies, 59000 Lille, France; anne.loyens@inserm.fr

**Keywords:** oxidative stress, mitochondrial dysfunction, mitochondrial antioxidant, superoxide dismutase 2, acetylation, mitophagy, cardiac hypertrophy, heart failure

## Abstract

Heart failure, mostly associated with cardiac hypertrophy, is a major cause of illness and death. Oxidative stress causes accumulation of reactive oxygen species (ROS), leading to mitochondrial dysfunction, suggesting that mitochondria-targeted therapies could be effective in this context. The purpose of this work was to determine whether mitochondria-targeted therapies could improve cardiac hypertrophy induced by mitochondrial ROS. We used neonatal (NCMs) and adult (ACMs) rat cardiomyocytes hypertrophied by isoproterenol (Iso) to induce mitochondrial ROS. A decreased interaction between sirtuin 3 and superoxide dismutase 2 (SOD2) induced SOD2 acetylation on lysine 68 and inactivation, leading to mitochondrial oxidative stress and dysfunction and hypertrophy after 24 h of Iso treatment. To counteract these mechanisms, we evaluated the impact of the mitochondria-targeted antioxidant mitoquinone (MitoQ). MitoQ decreased mitochondrial ROS and hypertrophy in Iso-treated NCMs and ACMs but altered mitochondrial structure and function by decreasing mitochondrial respiration and mitophagy. The same decrease in mitophagy was found in human cardiomyocytes but not in fibroblasts, suggesting a cardiomyocyte-specific deleterious effect of MitoQ. Our data showed the importance of mitochondrial oxidative stress in the development of cardiomyocyte hypertrophy. We observed that targeting mitochondria by MitoQ in cardiomyocytes impaired the metabolism through defective mitophagy, leading to accumulation of deficient mitochondria.

## 1. Introduction

Heart failure (HF) remains a major cause of illness and death and its prevalence is increasing, with a high rate of morbidity and mortality [1]. Despite major significant advances, HF remains a therapeutic challenge, and several adverse consequences of HF are still poorly controlled. The common phenotype associated with HF is the development of cardiac hypertrophy, defined as an increase in heart size in order to compensate for the increase in cardiac workload.

Oxidative stress, characterized by imbalanced reactive oxygen species (ROS) production and antioxidant defenses, plays an important role in regulating a wide variety of cellular functions, including gene expression, cell growth and death [2]. ROS cause contractile failure and structural damage in the myocardium [3] and activate a broad variety of prohypertrophic signaling kinases and transcription factors [4]. ROS production is well described as increasing in several animal models of cardiac diseases [2] such as cardiac alterations associated with obesity [5], myocardial infarction (MI) [6] or cardiomyocytes hypertrophy [7,8].

Among the several antioxidant defenses involved in biological systems protection from ROS toxicity, the superoxide dismutases (SOD) are metalloproteins playing an essential role by detoxifying the superoxide anions and preventing the formation of peroxynitrite. Three isoforms of SOD exist with specific subcellular localization: cytosolic SOD1, mitochondrial SOD2 and extracellular SOD3. We previously showed that inhibition of SOD2 expression induced both mitochondrial oxidative stress and cardiomyocytes hypertrophy [8], and mice deficient in SOD2 in cardiomyocytes have altered defective mitochondrial bioenergetics that cause lethal dilated cardiomyopathy [9]. Moreover, SOD2 activity is regulated by acetylation, mostly on the lysine 68 (K68) site, contributing to SOD2 inactivation [10] and to hypertension [11].

Although several sources contribute to oxidative stress, the majority of cardiac ROS come from mitochondria. Indeed, excessive ROS production in cardiac cells and tissues occurs during mitochondrial dysfunction, defined as decreased mitochondrial biogenesis, the number of mitochondria and altered membrane potential, becoming significant contributors to the development of cardiovascular disease [12]. Cardiac stress-induced mitophagy, the autophagy that selectively targets mitochondria, is triggered by a reduction in mitochondrial membrane potential, a metabolic stress or an accumulation of unfolded proteins [13,14]. Mitophagy is mainly regulated by PTEN-induced putative kinase protein-1 (PINK1)/parkin pathway, but also by FUN14 domain-containing 1 (FUNDC1) or Bcl2-interacting protein 3 (BNIP3) and BNIP3-like (BNIP3L/NIX) pathways [15]. In normal conditions, PINK1, a mitochondrial serine/threonine kinase, is maintained at low levels in the cardiomyocytes [16]. When mitophagy is active, PINK1 can phosphorylate the E3 ubiquitin ligase parkin at Ser65 [17], leading to parkin translocation from the cytoplasm to mitochondria, and then activation [18]. Parkin then ubiquinates several mitochondrial targets such as mitofusins or voltage-dependent anion channel 1 (VDAC1), which helps to remove damaged and dysfunctional mitochondria. Indeed, impaired mitochondrial function is associated with aging, MI and HF [15,19], suggesting that mitochondria-targeted therapies could be effective in HF [20,21].

In this context, mitoquinone (MitoQ), a derivative of coenzyme Q, has been demonstrated to effectively improve mitochondrial function and attenuate redox-related cardiomyopathies [5,22]. Nevertheless, some studies—notably in cancer cells—described that MitoQ could lead to ROS production, rapid membrane depolarization and apoptotic cell death [23,24,25]. In this context, the purpose of this work was (1) to characterize how mitochondrial oxidative stress is involved in cardiac hypertrophy, (2) to identify the mechanisms able to activate SOD2 by deacetylation during cardiac hypertrophy and (3) to determine if mitochondria-targeted therapies could improve cardiac phenotypes.

## 2. Materials and Methods

### 2.1. Animal Model

All animal experiments were performed according to the Guide for the Care and Use of Laboratory Animals published by the US National Institutes of Health (NIH publication NO1-OD-4-2-139, revised in 2011). Animals were used and experimental protocols performed under the supervision of a person authorized to perform experiments on live animals (F. Pinet: 59-350126 and E. Dubois-Deruy 59-350253). Approval was granted by the institutional ethics review board (CEEA Nord Pas-de-Calais N°242011, January 2012).

Before surgery, rats were anesthetized (sodium methohexital, 50 mg/kg intraperitoneal (IP)), while analgesia was administered before (xylazine 5 mg/kg IP) and 1 h after surgery (xylazine 50 mg/kg subcutaneously) as described [26]. MI was induced in 10-week-old male Wistar rats (Janvier, Le Genest St isle, France) by ligation of the left anterior descending coronary artery (*n* = 20 MI, *n* = 19 sham) [26,27]. Hemodynamic and echocardiographic measurements (Appendix A) were taken 2 months after surgery, followed by heart excision and plasma sampling, as previously described [26,28].

### 2.2. Cell Culture

#### 2.2.1. Primary Cultures of Neonatal Rat Cardiomyocytes and Fibroblasts

Primary cultures of neonatal rat cardiomyocytes (NCMs) and fibroblasts (NCFs) were prepared from heart ventricles of 1- or 2-day-old rats, as previously described [29,30]. Briefly, cardiac cells were dissociated by enzymatic digestion with 0.04% collagenase II (Worthington, Lakewood, NJ, USA) and 0.05% pancreatin (Sigma-Aldrich, St. Louis, MO, USA) at 37 °C. NCMs and NCFs were separated from cell suspension by centrifugation 30 min at 3000 g in a discontinuous Percoll gradient (bottom 58.5%, top 40.5%, Sigma-Aldrich).

NCMs were seeded at a density 4 or 8 × 10^5^ cells/well in 6-well plates coated with 0.01% of collagen (Sigma-Aldrich) and cultured in a medium containing DMEM/Medium199 (4:1), 10% horse serum (Thermo Fisher Scientific, Waltham, MA, USA), 5% fetal bovine serum (FBS) (LGC Standards, Teddington, UK), 1% penicillin and streptomycin (P/S) (10,000 U/mL, Thermo Fisher Scientific) at 37 °C under 5% CO_2_ atmosphere. NCMs were starved for 24 h before Isoproterenol (Iso, 10 µmol/L) or PBS (as control) treatment for 24 h. A pre-treatment with mitoquinone (MitoQ, 1 µmol/L, 2 h) was also used. For autophagy experiments, Bafilomycin A (Baf, 10 nmol/L) was added in co-treatment for 2 h with the pre-treatment with MitoQ and for 24 h with Iso treatment.

NCFs were seeded at a density 3.5 × 10^5^ cells/well in 6-well plates and cultured in a medium containing DMEM/Glutamax, 10% FBS, 1% P/S at 37 °C under 5% CO_2_ atmosphere. NCFs were starved for 24 h before pre-treatment of MitoQ (1 µmol/L, 2 h) or DMSO (as control) for 24 h in the privation medium.

#### 2.2.2. Transfection

The siRNA specifically targeting rat sirtuin 3 mRNA (si SIRT3: (ON-TARGETplus Rat Sirt3 siRNA—SMARTpool, # L-084761-03-0010), rat parkin mRNA (si prk: ON-TARGETplus Rat Prkn (56816) siRNA—SMARTpool, #L-090709-02-0010) and non-targeting control (si NT: ON-TARGETplus Non-targeting Control Pool, D-001810-10-20) were used. NCMs were plated (6.5 × 10^5^ cells/well) in 6-well plates and were allowed to grow for 24 h without P/S. siRNA (25 nmol/L) were transfected with the DharmaFECT^®^ reagent (4 µL) according to the manufacturer’s recommendations (Horizon Discovery, Cambridge, UK).

For SIRT3 sur-expression, the NCMs were transfected with the plasmid pCMV6 in which the nucleotide sequence of the SIRT3 protein (RN203062, OriGene, Rockville, MD, USA) was cloned. The empty pcMV6 plasmid (PS100001, OriGene) was used as a control. NCMs were plated (6.5 × 10^5^ cells/well) in 6-well plates and were allowed to grow for 24 h in complete medium. The plasmid (3 μg) with 3 μL of lipofectamine 2000 transfecting agent (11668019, Thermo Fisher Scientific) were added per well.

For both experiments, total cell extracts were collected 72 h after transfection.

#### 2.2.3. Primary Cultures of Adult Rat Cardiomyocytes

Adult rat cardiomyocytes (ACMs) were isolated as previously described [31]. Briefly, hearts were collected from male Wistar rats and perfused in Krebs–Henseleit buffer containing 10 mmol/L HEPES, 5 mmol/L glucose, 2 mmol/L pyruvate and 25 mmol/L NaCl. Hearts were then digested in Krebs–Henseleit buffer containing 1 mg/mL collagenase type II and 0.4% free fatty acid BSA for 30 min at 37 °C. After collagenase perfusion, hearts were removed from the perfusion apparatus, cut into small fragments and incubated under agitation for 10 min at 37 °C in Krebs–Henseleit Buffer containing 0.02 mmol/L CaCl_2_. CaCl_2_ was progressively added to the medium to reach 1 mmol/L final concentration. After sedimentation and washing, cells were finally resuspended in MEM medium containing 20 mmol/L HEPES, 2.5% FBS and 2% P/S and equally seeded in 3 cm diameter wells (20 wells for one digested heart) previously coated with laminin and incubated at 37 °C for at least 1 h. Two hours after seeding, ACMs were starved for 24 h before Iso (50 µmol/L) or PBS (as control) treatment for 48 h. A pre-treatment of MitoQ (0.25 to 5 µmol/L, 2 h) or DMSO (as control) were also used.

#### 2.2.4. Human Cardiomyocytes

Human Cardiac Myocytes (HCMs) (C-12810, PromoCell, Heidelberg, Germany) were seeded at a density 1.5 × 10^5^ cells/well in 12-well plates and cultured in a myocyte growth medium (C-22170, PromoCell) containing 0.5 ng/mL epidermal growth factor, 2 ng/mL basic fibroblast growth factor, 5 μg/mL insulin and 5% fetal calf serum. HCMs were starved for 24 h before pre-treatment of MitoQ (1 µmol/L, 2 h) or DMSO (as control). Cells were then maintained during 24 h in the privation medium.

### 2.3. Cell Index Quantification by Real Time Cell Analysis (RTCA)

NCMs were seeded in a 0.64 cm^2^ well covered with 80% gold electrodes (00300600840, Agilent Technologies, Santa Clara, CA, USA) from which a low-voltage alternating current (20 mV) was generated in the culture medium from one electrode to another by an iCELLingence system (00380601000, Agilent Technologies). The electrical resistance of adherent cells (RCell) was registered by a control unit (00380601430, Agilent Technologies) equipped with RTCA Lite software (00310100210, Agilent Technologies) and was expressed at each time in cellular index (ICell) defined by the equation: ICell = (RCell − Rm)/Rm where Rm corresponds to the resistance of the medium without the cells. The L8 E-Plates were incubated at 37 °C and 5% CO_2_ and the cell index was recorded every 15 min until the end of the experiment.

### 2.4. RNA Extraction and qRT-PCR Analyses

RNA was extracted from NCMs with QIAGEN RNeasy Mini Kit (Qiagen, Germantown, MD, USA), as described by the manufacturers’ instructions. Reverse transcription was performed with 100, 250 or 500 ng of total RNA using the miScript II RT kit (Qiagen) and the cDNA was amplified with miScript SYBR Green PCR (Qiagen) on an Aria Mx Q-PCR system (Agilent Technologies), according to the manufacturer’s instructions. The sequences of the different primers (Eurogentec, Seraing, Belgium) used were: mitochondrial fission 1 protein (*Fis 1*) (accession number: 288584; sense: GCACGCAGTTTGAATACGCC, antisense: CTGCTCCTCTTTGCTACCTTTGG); hypoxanthine phosphoribosyltransferase 1 (*HPRT*) (accession number: 24465; sense: ATGGGAGGCCATCACATTGT, antisense: ATGTAATCCAGCAGGTCAGCAA); mitofusin 2 (*Mfn2*) (accession number: 64476; sense: GATGTCACCACGGAGCTGGA, antisense: AGAGACGCTCACTCACTTTG); *NRF2* (accession number: 83619; sense: GCAACTCCAGAAGGAACAGG, antisense: AGGCATCTTGTTTGGGAATG) and peroxisome proliferator-activated receptor gamma coactivator 1-alpha (*PGC1α*) (accession number: 83516; sense: AAAAGCTTGACTGGCGTCAT, antisense: TCAGGAAGATCTGGGCAAAG). ΔΔCT method was used for data analysis.

### 2.5. Protein Extraction and Western Blot

#### 2.5.1. Protein Extraction

Proteins were extracted from 6-well plates cell or from 40 mg of LV frozen tissue with Dounce–Potter homogenization into ice-cold RIPA buffer (50 mmol/L Tris [pH 7.4], 150 mmol/L NaCl, 1% Igepal CA-630, 50 mmol/L deoxycholate and 0.1% SDS) containing antiproteases (Complete™ EDTA-free, Sigma-Aldrich), serine/threonine protein phosphatase inhibitors (Phosphatase inhibitor Cocktail 3, Sigma-Aldrich) and 1 mmol/L Na_3_VO_4_. Lysates were incubated for 1 h at 4 °C and centrifuged 15 min at 11,000× *g* to collect the soluble proteins. Protein concentrations were determined with a Lowry-based method protein assay (Bio-Rad, Hercules, CA, USA) and samples were kept at −80 °C.

#### 2.5.2. Cytosol-Mitochondria Fractionation

NCMs were seeded at 2 × 10^6^ cells/dish before Iso treatment as described above. Heart or cells were washed with 5 mL ice-cold PBS and dissociated in 750 µL of ice-cold buffer H (0.3 mol/L sucrose, 5 mmol/L TES, 2 mmol/L EGTA, pH 7.2) [32]. Scrapped cells or tissue were then transferred in a 2 mL tube in which 500 µL of buffer H containing 1 mg/mL BSA was added. After gently turning the tube 6 times from bottom to top, the tubes were centrifuged for 10 min at 500× *g* and the supernatant collected was centrifuged for 10 min at 3000× *g*. Cytoplasmic fraction (supernatant) was collected and 100 µL of RIPA 2× buffer added. Mitochondrial fraction (pellet) was resuspended in 50 µL RIPA 2× buffer. The purity of our fractionation was validated by detection of ATP synthase α only in the mitochondrial fraction.

#### 2.5.3. Cytoplasm-Nuclei Fractionation

NCMs were seeded at 4 × 10^5^ cells/dish before Iso treatment as described above. Cytoplasm–nuclei fractionation was performed with NE-PER™ Nuclear and Cytoplasmic Extraction Reagents (78835, Thermo Fisher Scientific) following manufacturer’s instructions. After treatment, media were removed and cells washed with 5 mL ice-cold PBS before trypsinization for 20 min at 37 °C. After trypsin inactivation, NCMs were centrifuged for 7 min at 800× *g*. Pellet was resuspended with 1 mL ice-cold PBS before centrifugation for 10 min at 3000× *g*. Pellet was then incubated with 100 µL of ice-cold CER I buffer for 10 min in ice. Next, 15 µL of ice-cold CER II buffer was then added and incubated on ice for 1 min. The tubes were then centrifuged for 5 min at 16,000× *g*. The supernatant (cytoplasmic extract) was collected and the pellet fraction, which contains nuclei, was resuspended in ice-cold NER buffer. Pellet was then vortexed on the highest setting for 15 s every 10 min for a total of 40 min. The tubes were then centrifuged for 10 min at 16,000× *g* and the supernatant (nuclear extract) fraction was collected. The purity of our fractionation was validated by detection of lamin B1 only in the nuclei fraction.

#### 2.5.4. Western Blot (WB)

Soluble proteins (10 to 50 µg) were resolved on NuPAGE 4–12% Bis-Tris Protein Gels (Thermo Fisher Scientific) or on SDS-PAGE gels (12 or 15%, depending on the proteins analyzed) and transferred on 0.2 μm nitrocellulose membranes (Trans-Blot^®^ TurboTM Transfert Pack, Bio-Rad). Equal total proteins loads were verified by Ponceau red (0.1% Ponceau, Sigma-Aldrich), 5% acetic acid (*v*/*v*) staining of the membranes. The membranes were then blocked in 5% milk or 5% BSA in TBS-Tween buffer for 1 h before 4 °C overnight incubation with primary antibodies diluted in blocking solution. Blots were then washed three times with TBS-Tween 0.1% buffer and incubated with corresponding secondary antibodies for 1 h (1/5000 to 1/10,000) in blocking solution. The Chemidoc^®^ camera (Bio-Rad) was used for imaging and densitometry analysis after membranes were incubated with enhanced chemiluminescence (ECL™) Western blotting detection reagents (GE Healthcare, Chicage, IL, USA).

### 2.6. Cell Staining

#### 2.6.1. Immunofluorescence (IF)

Biphotonic confocal microscopy was used for the imaging of 4% paraformaldehyde and 0.1% Triton fixed/permeabilized cardiomyocytes. Immunofluorescence staining was performed by saturation for 30 min with 1% BSA before incubation with primary antibodies overnight at 4 °C. Alexa Fluor^®^ secondary antibody (Thermo Fisher Scientific) at dilution 1/300 was incubated for 30 min at room temperature before nuclei staining for 10 min at room temperature (Hoechst 33258, Thermo Fisher Scientific) with mounting medium (Vectashield, Eurobio Scientific, Paris, France).

Cardiomyocytes were incubated in Hank’s Balanced Salt Solution (HBSS) with 10 μmol/L MitoSOX Red (Thermo Fisher Scientific) to stain mitochondrial superoxide anion levels and 100 nmol/L MitoTracker Deep red (M22426, Thermo Fisher Scientific) to stain mitochondria or 10 µg/mL of JC-1 sensor (T3168, Thermo Fisher Scientific) to quantify the mitochondrial membrane potential for 30 min at 37 °C.

Staining was visualized with an ×40 objective of LSM710 confocal microscope that used Zen image acquisition and analysis software (Zen 14.0.0.201) (Zeiss, Rueil Malmaison, France). Images were acquired with a resolution of at least 1024 × 1024 and analyzed with Image J software (version 1.48i, public domain software). Colocalization was quantified by Pearson’s coefficient, describing the correlation between the intensities of 2 proteins with the JACoP plugins on ImageJ software.

#### 2.6.2. Proximity Ligation Assay (PLA)

We performed PLA following manufacturer’s instructions (Olink Bioscience, Uppsala, Sweden). Briefly, after primary antibody incubation as described before, samples were incubated for 1 h at 37 °C with secondary antibody coupled to oligonucleotidic probes (Duolink^®^ In Situ PLA^®^ Probe Anti-Mouse PLUS and Anti-Rabbit MINUS, Sigma-Aldrich), diluted at 1/5 in BSA 1%. After 2 washes with PBS 1X, samples were incubated for 30 min at 37 °C with ligase (1/40 in ligation solution), then washed twice more with PBS 1X before incubation for 1 h 40 at 37 °C with polymerase (1/80 in amplification solution). After washing with PBS, coverslips were mounted with glycerol/PBS 10X (90/10, *v*/*v*).

Staining was visualized with an ×40 objective of LSM710 confocal microscope followed by Zen image acquisition. Images were acquired with a resolution of at least 1024 × 1024 and analyzed with Image J software. Mean fluorescence intensity and number of spots by cells was quantified by a “home-made” plugin write on Image J software.

#### 2.6.3. Detection of Mitochondrial Hydrogen Peroxide Levels Using MitoPY1

NCMs were seeded at 2 × 10^5^ cells/well in 96 well-plate. The cells were serum-deprived for 24 h and treated with PBS (control) or Iso at 10 µmol/L for 24 h in serum-free medium. After treatment, media was removed and cells were washed twice with 100 µL PBS 1X/well and then immediately stained with 50 µmol/L MitoPY1 probe (4428, Sigma-Aldrich) for 30 min at 37 °C in PBS 1X. After staining, cells were washed twice in 100 µL PBS 1X. Then, 200 µL PBS 1X was added in 96 well-plates and absorbance was measured with a microplate reader at an excitation wavelength of 485 nm and an emission wavelength of 520 nm.

#### 2.6.4. Transmission Electronic Microscopy

NCMs were fixed overnight in 2.5% glutaraldehyde in 0.1 mol/L phosphate buffer, and then washed 3 times with Phosphate buffer 0.1 mol/L. Samples were then post-fixed in 1% osmium tetroxide in Phosphate buffer 0.1 mol/L at room temperature for 1 h then followed by dehydration steps (5 min in ethanol 50%, 5 min in ethanol 70%, 5 min in ethanol 80%, 2 × 15 min in ethanol 95%, 3 × 20 min in ethanol 100%). Cells were centrifuged 10 min at 12,000 rpm. Pellets were then incubated with propylene oxide for 30 min. Samples were then stained in propylene oxide/Epon (*v*/*v*) for 1 h, then in Epon 100% twice for 1 h, followed by overnight incubation before capsules embedding at 60 °C for 4 days.

Ultrathin sections (85 nm) were performed with UM EC7 ultramicrotome (Leica, Wetzlar, Germany), and sections were contrasted by uranyl acetate 2%/ethanol 50% treatment for 8 min followed by Reynolds lead citrate for 8 min. Sections were observed using a EM900 electron microscope (Zeiss) with Gatan Orius SC1000 camera. Quantification of mitochondria number, length, width and area were performed with analysis software (Zen 2.3) (Zeiss).

### 2.7. Antibodies

The primary antibodies used were 3-nitrotyrosine (mouse, ab110282, Abcam, Cambridge, UK, 1/1000 WB); aconitase 2 (rabbit, GTX109736, CliniSciences, Nanterre, Paris, France, 1/10,000 WB); α-actinin antibody (mouse, A-7811, Sigma-Aldrich, 1/50 IF); ATP synthase alpha (mouse, A-21350, Thermo Fisher Scientific, 1/5000 WB); B-cell lymphoma 2 (Bcl-2) (mouse, sc-7382, Santa cruz, Dallas, TX, USA, 1/300 WB), beclin-1 (rabbit, #3738, Ozyme, Saint-Cyr-l’École, France 1/1000 WB); catalase (mouse, CO979, Sigma-Aldrich, 1/1000 WB); glyceraldehyde-3-phosphate dehydrogenase (GAPDH) (mouse, sc-365062, Santa Cruz, 1/10,000 WB); Lamin B1 (rabbit, ab133741, Abcam, 1/1000 WB); LC3 B (rabbit, #2775, Ozyme, 1/2000 WB and rabbit, NB600-1384, Bio-Techne, Minneapolis, MN, USA, 1/50 IF); parkin (rabbit, 702785, Thermo Fisher Scientific, 1/1000 WB and mouse, P6248, Sigma-Aldrich, 1/50 IF); peroxiredoxin-1 (Prx1) (rabbit, ab59538, Abcam, 1/1000 WB); sarcomeric-actin (mouse, M0874, Agilent Technologies, 1/5000 WB), SIRT1 (mouse, #8469, Ozyme, 1/1000 WB and rabbit, #9475, Ozyme, 1/50 IF); SIRT3 (rabbit, #5490, Ozyme, 1/1000 WB and 1/50 IF and PLA); SIRT6 (rabbit, ab191385, Abcam, 1/1000 WB), SOD1 (rabbit, 10269-1-AP, Proteintech, Manchester, UK, 1/1000 WB); SOD2 (rabbit, ab13533, Abcam, 1/5000 WB and mouse, ab16956, Abcam, 1/50 IF and PLA), SOD2 acetylated on lysine 68 (SOD2acK68) (rabbit, ab137037, Abcam, 1/1000 WB), SOD2 acetylated on lysine 122 (SOD2acK122) (rabbit, ab214675, Abcam, 1/1000 WB), ubiquitinylated proteins (mouse, BML-PW8810-0500, Enzo Life Science, Villeurbanne, France, 1/2500 WB).

The horseradish peroxidase-labeled secondary antibodies used for WB were Anti-Rabbit IgG (NA934V) and Anti-Mouse IgG (NA931) antibodies from GE healthcare.

The Alexa Fluor^®^ secondary antibodies used for IF were Alexa Fluor 488 Chicken Anti-Rabbit IgG (A21441, Thermo Fisher Scientific) and Alexa Fluor 568 Donkey Anti-Mouse IgG (A10037, Thermo Fisher Scientific).

### 2.8. Oxygraphy Analysis

NCMs were seeded at 4 × 10^5^ cells/well in 6 well-plates (1 plate per condition) in culture medium with serum. Then, cells were serum-deprived for 24 h and treated with PBS (control) or Iso at 10 µmol/L for 24 h and 48 h in serum-free medium. A pre-treatment of MitoQ (1 µmol/L, 2 h), or DMSO (as control) was also used. At the end of treatment, NCMs were gently rinsed twice with PBS. Pre-warmed trypsin solution was added into each well and incubated at 37 °C for 10 min. Once cells appeared detached, trypsin was inactivated by addition of medium containing serum and centrifuged at 800× *g* for 7 min. After removing the supernatant, the cell pellet was resuspended in pre-warmed medium with serum and count with Malassez cell.

NCMs (between 10^6^ and 1.5 × 10^6^ cells) were incubated into the O2K oxygraph chambers (Oroboros Instruments, Innsbruck, Austria) at 37 °C under constant stirring. After a 15 min stabilization leading to resting respiration, oligomycin A (5 nmol/L) was added to measure leak respiration for 5 min. Then, carbonyl cyanide m-chlorophenyl hydrazine (CCCP) pulses (0.5 to 2.5µmol/L steps) were performed until maximal oxygen consumption was achieved. Non-mitochondrial oxygen consumption was obtained after antimycin A (AA) (2.5 µmol/L) injection for 5 min.

### 2.9. Statistical Analysis

Data are expressed as medians with interquartile ranges and analyzed with GraphPad software version 7.0 (GraphPad, San Diego, CA, USA). Data were compared using non-parametric Mann–Whitney test. For RTCA experiments, the data were analyzed by functional ANOVA, performed using the *R* package fdANOVA [33]. Statistical significance was accepted at the level of *p* < 0.05.

## 3. Results

### 3.1. Characterization of Mitochondrial Oxidative Stress in Hypertrophied Neonatal Rat Cardiomyocytes

To understand the impact of oxidative stress on cardiac hypertrophy specifically in cardiomyocytes, we used the model of hypertrophied NCMs [34]. First, the cell index quantified by RTCA technology was used to determine the time point from which the NCMs profiles were stable. Indeed, the curve shows the dynamic change in the cell index, which represents a relative change in electrical impedance depending on the proliferation of the cultured cells (first exponential phase) (Appendix A). After serum privation, the curve stabilized and then the global cell index significantly decreased in hypertrophied NCMs (Iso) compared to control (PBS) (Figure 1a).

Based on these data, NCMs were Iso-treated during 24 h for the following experiments. We validated the development of hypertrophy in NCMs with a significant increase in cell area (Figure 1b) and of mitochondrial superoxide anion levels quantified with a MitoSOX probe (Figure 1c) without any mitochondrial hydrogen peroxide accumulation (Appendix A). We also quantified antioxidant enzymes and observed no significant modulation of catalase, Prx-1, SOD1 or SOD2 (Appendix A) following hypertrophy induction.

As acetylation contributes to SOD2 inactivation, we quantified the acetylated form of SOD2 on K68 or K122 and observed a significant increase in the SOD2acK68/SOD2 ratio, without modulation of SOD2acK122/SOD2 ratio (Figure 1d and Appendix A), meaning that SOD2 activity is significantly decreased in hypertrophied NCMs. We also confirmed that SOD2 and SOD2acK68 are both localized in mitochondria independently of Iso treatment by subcellular cytosol/mitochondria fractionation (Figure 1e) and colocalization with a MitoTracker probe (Figure 1f), with no significant impact of Iso treatment on the mitochondrial localization of SOD2.

As several SIRTs have been described to deacetylate/activate SOD2, we quantified SIRT1 [35], SIRT3 [36] and SIRT6 [37] expression in hypertrophied NCMs and observed no significant modulation of SIRT1, SIRT3 and SIRT6 (Appendix A). Moreover, we investigated a possible interaction between SOD2 and one or several SIRTs. Through subcellular cytosol/mitochondria (Figure 1e) and cytoplasm/nuclei (Appendix A) fractionation, we observed that SIRT1 and SIRT3 are localized in mitochondria with SOD2 and SOD2acK68 whereas SIRT6 is strictly expressed in nuclei.

Interestingly, colocalization with a MitoTracker probe showed a significant decrease in the mitochondrial localization of SIRT3 with Iso treatment (0.83 [0.80–0.86] PSB vs. 0.69 [0.60–0.79] Iso, *p* = 0.0140) (Figure 1g) whereas no impact of Iso treatment on the mitochondrial localization of SIRT1 was observed (Appendix A). We validated the interaction between SIRT3 and SOD2 by PLA (Appendix A) and observed a significant decrease in mean intensity and number of spots due to hypertrophy as well as a significant decrease in mitochondrial localization of PLA signal (Appendix A). These results indicate that a decrease of SIRT3 in mitochondria might be involved in the increase of SOD2acK68 observed in Iso-treated NCMs.

### 3.2. Modulation of SOD2 by SIRT3 Impacts Mitochondrial Oxidative Stress in Hypertrophied Neonatal Rat Cardiomyocytes

To further study the impact of SIRT3-mediated SOD2 activation in the cardiomyocytes, we first inhibited SIRT3 expression using siRNA in untreated and Iso-treated NCMs. SIRT3 siRNA induced a 50% and 70% decrease in SIRT3 expression in control (PBS) and hypertrophied (Iso) (Figure 2a and Table 1) NCMs, respectively. No modulation of SIRT1 was observed after SIRT3 silencing (Figure 2a and Table 1), suggesting that SIRT3 inhibition is not compensated by overexpression of SIRT isoforms. The silencing of SIRT3 only induced a significant decrease in SOD2 expression in hypertrophied NCMs (Figure 2b and Table 1).

We then observed that SIRT3 inhibition was associated with a significant increase in superoxide anion production in hypertrophied NCM after MitoSOX staining (Figure 2c) associated with a significant decrease in cardiomyocytes area (Figure 2d) and no significant modulation of the antiapoptotic Bcl2 protein (Table 1). Altogether, these results showed the deleterious effect of SIRT3 inhibition on Iso-induced hypertrophy and oxidative stress by inhibition of SOD2 expression.

In the other hand, the overexpression of SIRT3 in NCMs led to a two-fold increase in its expression in NCMs control (PBS) and hypertrophied (Iso) was associated with increased SIRT1 expression (Figure 3a and Table 2), suggesting the possibility of cross-regulation of the different SIRTs isoforms expressions in the heart. We also observed a significant decrease in the SOD2acK68/SOD2 ratio in NCMs control (PBS) and hypertrophied (Iso) without modulation of SOD2 expression (Figure 3b and Table 2). Moreover, SIRT3 overexpression was associated with a significant decrease in mitochondrial superoxide anion production (Figure 3c) as well as of cardiomyocytes area (Figure 3d) in Iso-treated NCMs. These results suggest that SIRT3 overexpression leads to an increase in SOD2 activity, which is shown by decreased SOD2acK68 expression to protect NCMs from Iso-induced hypertrophy and mitochondrial oxidative stress. This seems validated by the trend to increase the antiapoptotic Bcl2 protein (Table 2).

### 3.3. SIRT3 Regulates SOD2 Deacetylation in Ischemic Heart In Vivo

We then used the well-described in vivo model of HF-rats induced by coronary ligation [26,38] to further investigate the role of SIRT3-mediated SOD2 deacetylation in the heart. In this model, we previously showed an increased SOD2 expression in the left ventricle (LV) of HF-rats 2 months after MI [8]. Here, we observed a significant decrease in its acetylated form (SOD2acK68/SOD2) in LV of HF-rats, suggesting a potential increase in SOD2 activity and a trend to an increase in SIRT3 expression (Appendix A). By subcellular cytosol/mitochondria fractionation, we confirmed a potential interaction between SOD2 and SIRT3 with a mitochondrial localization of SOD2acK68, SOD2 and SIRT3 (Appendix A). Moreover, the protein nitro-oxidation (quantified with 3-nitrotyrosine antibody) was not modulated in HF-rats (Appendix A). These data suggest that SIRT3 could promote active SOD2 expression in the heart to prevent from oxidative stress.

To confirm the involvement of SIRT3 in SOD2 deacetylation in vivo, we then characterized adult cardiomyocytes isolated from SIRT3 KO mice. A total abolition of SIRT3 expression and a significant increase in SOD2acK68 were observed in cardiomyocytes of SIRT3 KO mice (Appendix A), validating the involvement of SIRT3 in regulation of SOD2 deacetylation in vivo in the heart.

### 3.4. Characterization of Mitochondrial Biogenesis and Mito(auto)phagy in Hypertrophied Neonatal Rat Cardiomyocytes

As SOD2 is a mitochondrial enzyme, we investigated mitochondrial function and quantified mitochondrial respiration by oxygraphy using isolated NCMs treated (or not) with Iso. We observed a classical profile, including basal respiration, ATP production-coupled respiration (oligomycin A), maximal and reserve capacities (CCCP) and non-mitochondrial respiration (AA), but we did not observe any significant modulation of oxygen consumption upon Iso treatment for 24 h compared to untreated NCMs (Appendix A). Interestingly, we observed a trend demonstrating a decrease in maximal oxygen consumption after 48 h of Iso (Appendix A).

We then investigated the main regulator of mitochondrial biogenesis, *PGC1α*, and one of its transcriptional coactivators, *NRF2*, as well as different genes involved in mitochondrial fusion (*Mfn2*) and fission (*Fis1*) in hypertrophied NCMs. We observed a significant decrease in mitochondrial biogenesis quantified by the decreased mRNA levels of *PGC1α*, *NRF2*, *Mfn2* and *Fis 1* (Table 3). These results suggested that mitochondrial biogenesis is impaired during hypertrophy, which is corroborated by electronic microscopy showing a significant decrease in the number of mitochondria, but these are altered (loss of electron-dense matrix) and larger (significant increased length, width and area) (Figure 4a). Moreover, we also observed a significant decrease in aconitase 2, an enzyme of the Krebs cycle (Table 3).

We then investigated how hypertrophy could affect mitophagy, an essential process for clearing away the defective mitochondria. As the inner mitochondrial membrane depolarization is the precursor step for mitophagy, we first used the JC-1 dye with a green fluorescent emission for the monomeric form of the probe and a red fluorescent emission for a concentration-dependent formation of J-aggregates. We observed a significant mitochondrial depolarization, indicated by the decreased red/green ratio, in hypertrophied NCMs (Figure 4b). We then quantified the proteins involved in the mitophagy/autophagy process and observed that Iso treatment induced a decrease in mitophagy/autophagy with a significant decrease in parkin and LC3II/LC3I ratio quantified by Western blotting (Figure 4c and Table 3) and immunofluorescence (Figure 4d). No modulation of ubiquitinated proteins and beclin-1 was observed (Figure 4c and Table 3). To determine if autophagy is active, NCMs were treated by Bafilomycin A1 (Baf) to inhibit the autophagosome–lysosome fusion. This inhibition induced an increase in the LC3II/LC3I ratio, but not of parkin, showing that autophagy is decreased but still active in hypertrophied NCMs (Appendix A).

### 3.5. Effect of Mitochondrial Antioxidant (MitoQ) on Oxidative Stress and Mitochondrial Biogenesis in Hypertrophied Neonatal Rat Cardiomyocytes

In this context, we pre-treated NCMs with MitoQ, a derivative of coenzyme Q that targets ROS in mitochondria. First, we observed two phases in the cell index of NCMs pre-treated by MitoQ with an increase during the first 2 h, corresponding to the pre-treatment followed by a progressive but significant decrease in cell index during the next 24 h (Figure 5a). This decrease was observed independently of the concentration of MitoQ used (from 0.25 to 5 µmol/L) and could be related to NCMs mortality (Appendix A).

As expected, we quantified a significant decrease in cardiomyocytes area as well as of superoxide anion production in NCM pre-treated by MitoQ before Iso treatment (Figure 5b,c). Interestingly, we observed a significant decrease in both SOD2acK68/SOD2 ratio and SIRT3 expression (Figure 5d) that could reflect impaired mitochondrial function. Indeed, the pre-treatment did not affect SOD2 mitochondrial localization, as confirmed by colocalization with the MitoTracker probe, but we observed that the mitochondria are altered in NCMs pre-treated by MitoQ (Appendix A). We also quantified the other antioxidant enzymes and observed no modulation of catalase but a significant decrease in Prx-1 and SOD1 in NCMs pre-treated by MitoQ (Appendix A), reflecting cellular stress.

In order to understand how MitoQ impaired the metabolism of cardiomyocytes, we investigated the mitochondrial biogenesis and respiration in hypertrophied NCMs pre-treated with MitoQ. Interestingly, we quantified a significant decrease of aconitase 2, an enzyme of the Krebs cycle, and *PGC1α* mRNA levels only in control NCMs pre-treated by MitoQ, whereas MitoQ induced a significant increase in *NRF2* mRNA levels in controls and hypertrophied NCMs pre-treated by MitoQ (Table 3). Moreover, mRNA levels of *Mfn2* and *Fis1* were not modulated (Table 3). Finally, we quantified mitochondrial respiration and observed a significant decrease in oxygen consumption with a loss of respiration capacity (shown by no response to CCCP) in hypertrophied NCMs pre-treated with MitoQ (Figure 5e). Moreover, electronic microscopy showed an aggravation of mitochondria alteration after pre-treatment with MitoQ, as indicated by the increased mitochondria area, length and width associated with an important loss of electron-dense matrix (Figure 5f). These results showed that mitochondria biogenesis and function are highly altered by MitoQ, independently of Iso.

### 3.6. Cardiomyocyte Specificity of the Detrimental Effect of Mitochondrial Antioxidant (MitoQ) on Mitophagy in Hypertrophied Rat Cardiomyocytes

As mitophagy is the selective pathway of degradation of defective mitochondria, we investigated how MitoQ could affect this process. We first used the JC-1 dye and quantified a significant and stronger decrease in the red/green ratio in hypertrophied NCMs pre-treated with MitoQ (Figure 6a). We then quantified the proteins involved in the mitophagy/autophagy process. Interestingly, through immunofluorescence (Figure 6b) or Western blotting (Figure 6c and Table 3) we observed the complete suppression of parkin expression by MitoQ (alone or with Iso). MitoQ pre-treatment significantly decreased the expression of beclin-1 and increased the LC3II/I ratio without modulation of ubiquitinated proteins (Figure 6b,c and Table 3), suggesting a dysregulation of mitophagy/autophagy induced by MitoQ. To determine if autophagy is active, NCMs were treated with Bafilomycin at the same time as MitoQ and Iso. The inhibition of the autophagosome–lysosome fusion did not modulate the LC3II/LC3I ratio, suggesting that autophagy is inactive in hypertrophied NCMs pre-treated with MitoQ (Appendix A).

We then quantified a significant and progressive decrease in superoxide anion production in hypertrophied adult cardiomyocytes (ACMs) pre-treated by MitoQ before Iso (Figure 6d). Moreover, we observed in parallel a constant decrease in cell number, probably due to stronger cell mortality (Figure 6d). These data showed the same effect of MitoQ in neonatal and adult rat cardiomyocytes. Interestingly, we did not observe any modulation of mitophagy/autophagy proteins in cardiac fibroblasts (NCFs) (Appendix A), suggesting that the impact of MitoQ on mitophagy is specific to cardiomyocytes. We then quantified the mitophagy/autophagy proteins in human cardiomyocytes (HCMs) and observed a significant decrease in parkin in HCMs pre-treated by MitoQ (Figure 6e) without modulation of other proteins involved in autophagy (ubiquitinated proteins and beclin-1). Surprisingly, we observed a single band for LC3, probably LC3I, which is significantly increased in HCMs pre-treated by MitoQ, as observed in NCMs (Figure 6e). These data validated the key role of parkin in cardiomyocytes to prevent oxidative stress. Indeed, the abolished expression of parkin upon MitoQ treatment led to defective mitophagy and accumulation of deficient mitochondria in cardiomyocytes.

### 3.7. Effect of Parkin Depletion on Hypertrophy, Oxidative Stress and Mitophagy in Neonatal Rat Cardiomyocytes

To understand how the decrease in parkin induced by MitoQ could affect cardiomyocytes, we transfected NCMs with the siRNA specifically targeting rat parkin mRNA (si prk). First, the cell index quantified by RTCA technology was non-significantly decreased in NCMs transfected with si prk compared to the non-target (si NT) (Figure 7a). The depletion of parkin did not impact the morphology of NCMs with no modulation of cell area (Figure 7b), but induced a significant decrease in mitochondrial superoxide anion production (Figure 7c). This decrease could be explained by a significant alteration of mitophagy reflected by a significant mitochondrial depolarization shown by the decreased red/green ratio in NCMs silenced for parkin (Figure 7d). The quantification of the proteins involved in the mitophagy/autophagy process showed that parkin expression was completely abolished by siRNA but LC3II/LC3I ratios were significantly increased without modulation of ubiquitinated proteins and beclin-1 (Figure 7e), suggesting a dysregulation of mitophagy/autophagy induced by parkin abolition.

## 4. Discussion

In this paper, we characterized how mitochondrial oxidative stress is involved in cardiac hypertrophy and how mitochondria-targeted therapies act to prevent these effects. We also highlighted the key role of mitophagy in the deleterious effect of mitochondrial antioxidant (MitoQ) in mitochondrial biogenesis despite its beneficial effect by reducing mitochondrial oxidative stress and cardiomyocytes hypertrophy.

Oxidative stress is considered a major regulator of the signal transduction in cardiac cells under pathological conditions. An understanding of the pathophysiological mechanisms which are involved in cardiac hypertrophy and the remodeling process is crucial for the development of new therapeutic strategies [4]. Here, we used an in vitro model of hypertrophied cardiomyocytes [34] to decipher the impact of oxidative stress. We observed an increase in mitochondrial oxidative stress associated with mitochondrial dysfunction. Indeed, excessive ROS production with mitochondrial dysfunction has been described as inducing irreversible damage to mitochondria, leading to the development of cardiovascular diseases [12]. For example, an increase in mitochondrial ROS production has been described in a murine model of MI induced by 4 weeks of coronary ligation [39] and ROS-generated by angiotensin II stimulation induced mitochondrial dysfunction, cardiomyocytes hypertrophy and HF [7]. Moreover, mitophagy, the selective autophagic removal of mitochondria, is essential for clearing away the defective mitochondria but can also lead to cell damage and death if excessive. At cardiovascular levels, mitophagy is involved in metabolic activity, cell differentiation, apoptosis and other physiological processes as reviewed recently [18].

SODs are metalloproteins capable of catalyzing the transformation of superoxide anion into hydrogen peroxide. This is the most effective antioxidant enzyme in humans [2]. Here, we showed that decreases in SOD2 activity are due to acetylation on K68 and correlate with mitochondrial ROS production, as previously described in other models such as human embryonic kidney cells [10] or hypertension [11,40] and with cardiomyocytes hypertrophy. Our data are consistent with our previous observations, showing that SOD2 inhibition induced both mitochondrial ROS production and hypertrophy of cardiomyoblasts [8]. Moreover, fine particulate matter exposure accelerated the risk of developing cardiac fibrosis and increased both mitochondrial ROS production and SOD2 acetylation [41]. Mice deficient in SOD2 died of cardiomyopathy within 10 days of birth, whereas heterozygous SOD2 (+/−) mice showed ultrastructural damage to the myocardium associated with increased oxidative stress [42]. More recently, a cardiomyocyte-specific SOD2 deficient mouse strain (SOD2Δ mice) was generated and, contrary to global SOD2 knockout mice, SOD2Δ mice died at 4 months due to HF [9]. Interestingly, these mice showed dilated cardiomyopathy as well as increased ROS production and mitochondrial dysfunction, confirming the association between SOD2 inactivation, ROS production and cardiac hypertrophy. In the in vivo model, we observed an activation of SOD2, from a decrease in acetylation, after 2 months of coronary ligation, which could be due to a late activation of SOD2. Indeed, we previously described an increase in mitochondrial superoxide anion after 24 h of Iso in cardiomyoblasts H9c2, whereas the mitochondrial superoxide anion decreased and SOD2 expression increased after 48 h, suggesting that SOD2 could be activated after long-term hypertrophy [8]. However, we could not discriminate the cell types responsible for SOD2 expression and activation in total left ventricle extract and could not eliminate a potential role of fibroblasts in in vivo results. Moreover, we demonstrate here that the K68 acetylation site is the one involved in cardiac SOD2 inactivation leading to mitochondrial superoxide anion production and hypertrophy in NCMs, and that SOD2 activation by deacetylation could be involved in HF post-MI.

Several SIRTs have been described to deacetylate/activate SOD2 in the heart, including SIRT1 [35], SIRT3 [36] and SIRT6 [37]. Due to localization and expression, we excluded the involvement of SIRT1 and 6 in SOD2 regulation in our model. Moreover, SIRT6 was previously described as directly binding and acetylating SOD2 on K68 and K122 in mouse hippocampal neuronal human and human embryonic kidney cell lines, suggesting a regulation of SOD2 acetylation dependent of the cell types [37]. Our results suggest that SIRT3 increases SOD2 activity, notably by decreasing its acetylation on K68, to protect NCMs from Iso-induced hypertrophy and mitochondrial oxidative stress. These data are in concordance with the previous result showing that SIRT3 inhibition in H9c2 cardiomyoblasts increased acetylated SOD2 and increased mitochondrial ROS production and cell death [43]. In NCMs, we observed that SIRT3 inhibition increased mitochondrial ROS production but decreased cell area; this might be explained by increased cell death. Indeed, SIRT3 overexpression decreased mitochondrial ROS production as well as cell area and tended to increase the antiapoptotic Bcl2 protein. Moreover, SIRT3 inhibition also seems to blunt mitophagy under hypoxia in cardiomyoblasts [43]. Overexpression of SIRT3 increased SOD2 activity, attenuated ROS production and improved mitochondrial bioenergetics in doxorubicin-treated H9c2 cardiomyoblasts, suggesting that SIRT3 activation could be a potential therapy for some cardiac dysfunction [36]. Furthermore, we also observed that SIRT3 inhibition decreases SOD2 expression that might be explained by an indirect transcriptional regulation of SOD2 by SIRT3 involving FOXO3A [44,45]. Our in vivo data indicate that SIRT3 is involved in regulation of SOD2 deacetylation in HF. Interestingly, mice subjected to transverse aortic constriction (TAC) surgery for 4 weeks developed cardiac hypertrophy and fibrosis associated with an increase in oxidative stress related to decreased SOD2 activity and SIRT3 expression [44].

It was suggested that mitochondria-targeted therapies could be effective in HF [20,21]. In this context, we selected the mitochondrial derivative of coenzyme Q (MitoQ), which decreased Iso-induced hypertrophy and mitochondrial oxidative stress. Surprisingly, we observed a deleterious effect of MitoQ on mitochondrial function and mitophagy. Controversial data described a protective or deleterious role of MitoQ. Indeed, in vitro, MitoQ has been described to prevent oxidative stress and alterations in mitochondrial proteins observed in palmitic acid-stimulated cardiomyoblasts [5]. In vivo, MitoQ reduced cardiac oxidative stress and prevented the development of cardiac fibrosis and hypertrophy in obese rats [5]. MitoQ could also significantly improve left ventricular dysfunction and increased metabolism-related gene expression in mice subjected to ascending aortic constriction [22]. These discrepancies could be explained by the cell specificity. Here, we specifically observed a decrease in parkin expression in the human and rat cardiomyocytes but not in the fibroblasts. Nevertheless, according to our data, some studies, notably in cancer cells, described that MitoQ could lead to ROS production, rapid membrane depolarization and apoptotic cell death [23,24]. Indeed, MitoQ seems to induce mitochondrial swelling and depolarization in kidney proximal tubule cells by a mechanism non-related to the antioxidant activity, but most likely because of the increased inner mitochondrial membrane permeability due to insertion of the alkyl chain [25]. These side effects of MitoQ are in accordance with our data showing impaired cardiomyocytes respiration, in link with the mitochondria ultrastructure and mitochondrial membrane potential alterations. The more striking point is the abolished expression of parkin upon MitoQ treatment leading to defective mitophagy and accumulation of deficient mitochondria in cardiomyocytes. We confirmed the key role of parkin through its silencing in cardiomyocytes, showing a defect in mitochondrial biogenesis and function.

## 5. Conclusions

In this paper, we identified that decreased interaction between SIRT3 and SOD2 in mitochondria in hypertrophied cardiomyocytes induced SOD2 inactivation associated with mitochondrial oxidative stress and dysfunction. We characterized how mitochondria-targeted therapies act to prevent these effects. We also highlighted the key role of mitophagy, and particularly of parkin, in the deleterious effect of mitochondrial antioxidant (MitoQ) in mitochondrial biogenesis, despite its beneficial effect of reducing mitochondrial oxidative stress and cardiomyocytes hypertrophy. The new hypothesis related to this work is that MitoQ improves cardiac hypertrophy and ROS production in cardiomyocytes but MitoQ induces impaired mitochondria morphology, oxygen respiration and membrane potential through a defect in mitophagy, specifically in cardiomyocytes. In conclusion, antioxidant therapeutic strategies should take into account the functional interaction between mitochondrial dynamism, biogenesis and mitophagy [46].

## Figures and Tables

**Figure 1 antioxidants-11-00723-f001:**
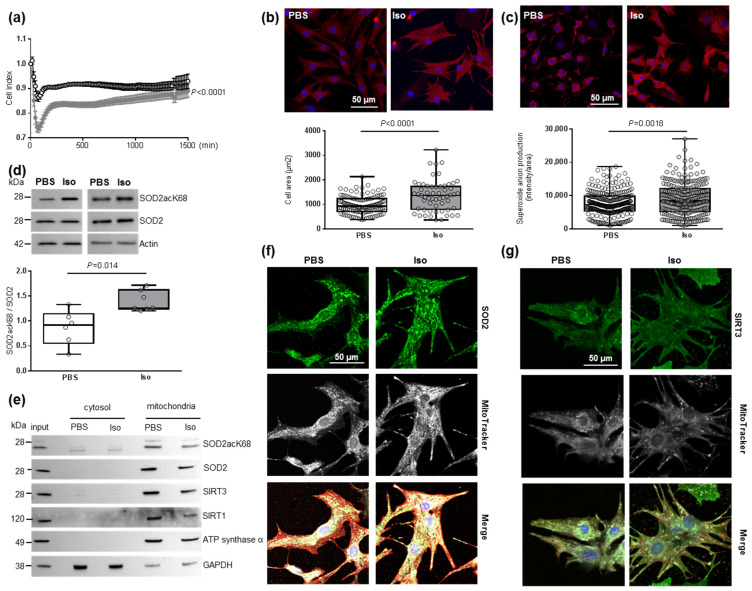
Characterization of mitochondrial oxidative stress in hypertrophied neonatal rat cardiomyocytes (NCMs). (**a**) Cell index quantification in untreated (black line) and Isoproterenol (Iso)-treated (gray line) NCMs by RTCA analysis. Cell index was recorded every 15 min (*n* = 4 independent isolation, in duplicate). (**b**) Hypertrophy was quantified in untreated- (PBS) (white box) or Iso-treated (gray box) NCMs for 24 h by immunofluorescence of alpha-actinin (red) and nuclei (blue) (top panels) and quantification of cell area (µm^2^) (bottom panel) (from 3 independent experiments and at least 273 cells). Oxidative stress was quantified in untreated (PBS) or Iso-treated NCMs for 24 h by fluorescence quantification of (**c**) mitochondrial superoxide anion with mitoSOX probes (red) (from 3 independent experiments and at least 236 cells) and by Western blot of (**d**) SOD2 acetylated on lysine 68 (SOD2acK68) or lysine 122 (SOD2acK122) on SOD2 ratio normalized to actin. (**e**) Representative images of SOD2acK68, SOD2 and sirtuins localized in mitochondria (validated by ATP synthase α) after subcellular fractionation of untreated (PBS) or Iso-treated-NCMs for 24 h. Representative images of (**f**) SOD2 (green) and (**g**) SIRT3 (green) localized in mitochondria (white) of untreated (PBS) or Iso-treated NCMs for 24 h. Colocalization appeared in merged images. Only significant *p* values are indicated from at least 3 independent experiments. Images were selected to represent the mean values of each condition.

**Figure 2 antioxidants-11-00723-f002:**
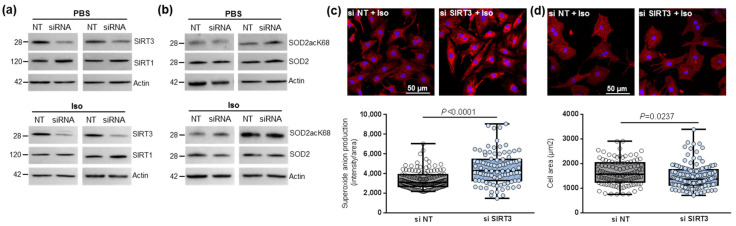
Effect of SIRT3 inhibition on SOD2 activation and mitochondrial oxidative stress in control and hypertrophied neonatal rat cardiomyocytes (NCMs). Control (PBS) or Iso-treated NCM were transfected without (gray box) or with (light blue box) SIRT3 siRNA for 72 h. Quantification by Western blot of SIRT3 and SIRT1 expression (**a**) and of SOD2acK68/SOD2 ratio and SOD2 expression (**b**) in Iso-treated NCM transfected with control or SIRT3 siRNA. Data were normalized on actin. (**c**) Mitochondrial superoxide anions production was detected after MitoSOX (red) and nuclei (blue) staining and quantified from 3 independent experiments and at least 92 cells. (**d**) Cell area was quantified from 3 independent experiments and at least 108 cells after alpha-actinin (red) and nucleus (blue) staining. Scale bar = 20 µm. Only significant *p* values are indicated from 3 independent experiments. Images were selected to represent the mean values of each condition.

**Figure 3 antioxidants-11-00723-f003:**
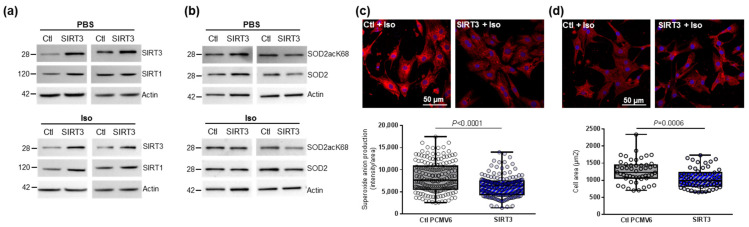
Effect of SIRT3 overexpression on SOD2 activation and mitochondrial oxidative stress in control and hypertrophied neonatal rat cardiomyocytes (NCMs). Control (PBS) or Iso-treated NCM were transfected with an empty vector (PCMV6) (gray box) or SIRT3 cDNA (3 µg, dark blue box) for 72 h. Quantification by Western blot of SIRT3 and SIRT1 expression (**a**) and SOD2acK68/SOD2 ratio and SOD2 expression (**b**) in Iso-treated NCM transfected with control or with SIRT3. Data were normalized on actin. (**c**) Mitochondrial superoxide anions production was detected after MitoSOX (red) and nuclei (blue) staining and quantified from 3 independent experiments and at least 168 cells. (**d**) Cell area was quantified from 3 independent experiments and at least 51 cells after alpha-actinin (red) and nucleus (blue) staining. Scale bar = 20 µm. Only significant *p* values are indicated from 3 independent experiments. Images were selected to represent the mean values of each condition.

**Figure 4 antioxidants-11-00723-f004:**
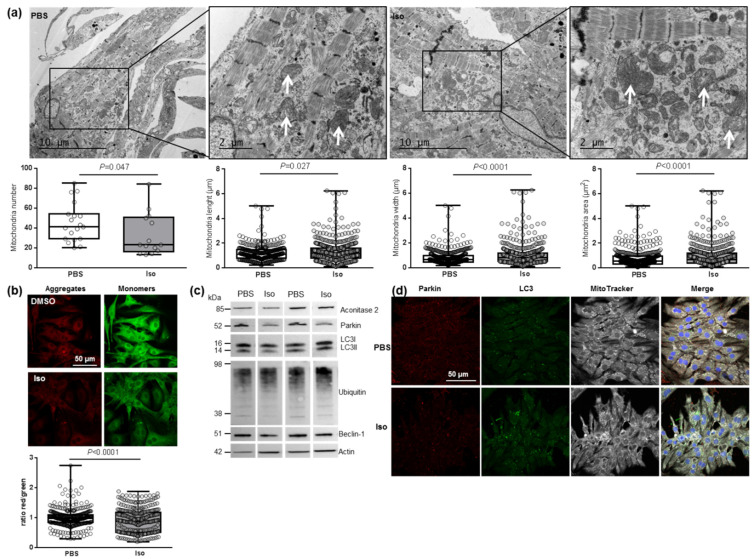
Characterization of mitochondrial function and autophagy/mitophagy in hypertrophied neonatal rat cardiomyocytes (NCMs). (**a**) Ultrastructure of PBS- and Iso-treated NCMs (magnification ×7000 with a scale bar of 10 µm (left images) and ×12,000 with a scale bar of 2 µm (right images)) and quantification of mitochondria number, length (µm), width (µm) and area (µm^2^). Arrows indicate example of mitochondria for comparison. (**b**) Mitochondrial membrane potential was quantified in NCMs treated with Iso by fluorescence quantification of JC-1 dye for aggregates (red) and monomer (green) (from 3 independent experiments and at least 366 cells). (**c**) Representative images for quantification of Krebs cycle by Western blot of aconitase 2 and mitophagy/autophagy by Western blot of parkin, LC3II/LC3I ratio, ubiquitin and beclin-1 in NCMs treated with Iso Data were normalized to actin. (**d**) Representative images of parkin (red) and LC3 (green) localized in mitochondria (white) of untreated (PBS) and Iso-treated NCMs. Colocalization appeared in merge images. Nuclei were stained by Dapi (blue). Only significant *p* values are indicated from at least 3 independent experiments. Images were selected to represent the mean values of each condition.

**Figure 5 antioxidants-11-00723-f005:**
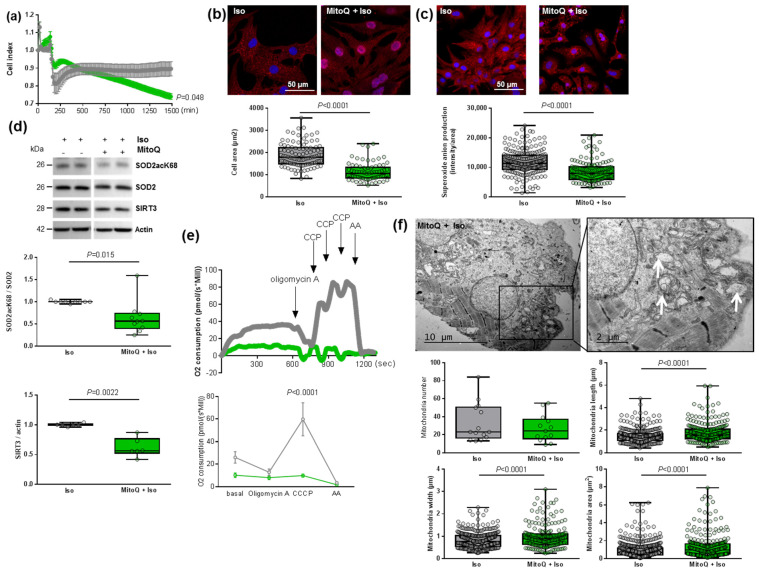
Effect of MitoQ in hypertrophied neonatal rat cardiomyocytes (NCMs). (**a**) Cell index quantification of hypertrophied NCM with pre-treatment with mitoquinone (MitoQ) (green line) or DMSO (as control) (gray line) by RTCA analysis. Cell index was recorded every 15 min (*n* = 4 independent isolation, in duplicate). (**b**) Hypertrophy was quantified in Iso-treated NCMs for 24 h with or without MitoQ by immunofluorescence of alpha-actinin (red) and nuclei (blue) (top panels) and quantification of cell area (µm^2^) (bottom panel) (from 3 independent experiments and at least 75 cells). (**c**) Oxidative stress was quantified in Iso-treated NCMs for 24 h with (+) or without (−) MitoQ by mitoSOX probe (red) (from 3 independent experiments and at least 133 cells). Quantification by Western blot of (**d**) SOD2 acetylated on lysine 68 (SOD2acK68) on SOD2 ratio and SIRT3 in Iso-treated NCMs for 24 h with or without MitoQ. Data were normalized to actin. (**e**) Representative oxygen consumption at basal level and after oligomycin, carbonyl cyanide m-chlorophenyl hydrazine (CCCP) and antimycin A (AA) addition (left panel) with quantification to characterize mitochondrial respiration (*n* = 9) (right panel) in Iso-treated NCMs with (green) or without (gray) MitoQ. (**f**) Ultrastructure of Iso-treated NCMs with or without MitoQ (magnification ×7000 with a scale bar of 10 µm (left images) and ×12,000 with a scale bar of 2 µm (right images)) and quantification of mitochondria number, length (µm), width (µm) and area (µm^2^). Arrows indicate examples of mitochondria for comparison. Only significant *p* values are indicated from at least 3 independent experiments. Images were selected to represent the mean values of each condition.

**Figure 6 antioxidants-11-00723-f006:**
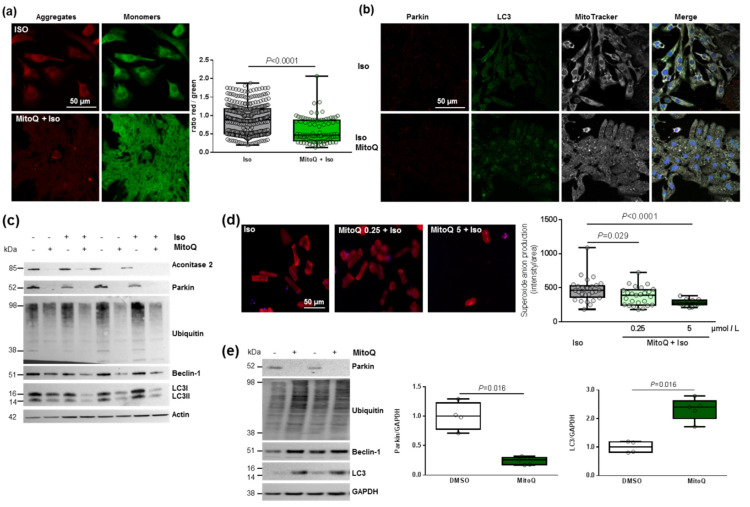
Effect of MitoQ on mitophagy in cardiomyocytes. (**a**) Mitochondrial membrane potential was quantified in hypertrophied NCM with pre-treatment with mitoquinone (MitoQ) by fluorescence quantification of JC-1 dye for aggregates (red) and monomer (green) (from 3 independent experiments and at least 94 cells). (**b**) Representative images of parkin (red) and LC3 (green) localized in mitochondria (white) of Iso-treated NCMs with pre-treatment with mitoquinone (MitoQ). Colocalization appeared in merge images. Nuclei were stained by Dapi (blue). (**c**) Mitophagy was quantified in control (−) or Iso-treated NCMs for 24 h (+) with (+) or without (−) MitoQ by Western blot of parkin, ubiquitinated proteins, beclin-1 and LC3II/LC3I ratio. Data were normalized to actin. (**d**) Mitochondrial superoxide anion was quantified in adult cardiomyocytes (ACMs) treated with Iso for 48 h with or without mitoQ pre-treatment (0.25 and 5 µmol/L) by fluorescence quantification of mitoSOX (red) (from at least 12 cells). (**e**) Mitophagy was quantified in human cardiomyocytes (HCMs) with (+) or without (−) MitoQ pre-treatment by Western blot of parkin, ubiquitinated proteins, beclin-1 and LC3. Data were normalized to GAPDH. Only significant *p* values are indicated from at least 3 independent experiments. Images were selected to represent the mean values of each condition.

**Figure 7 antioxidants-11-00723-f007:**
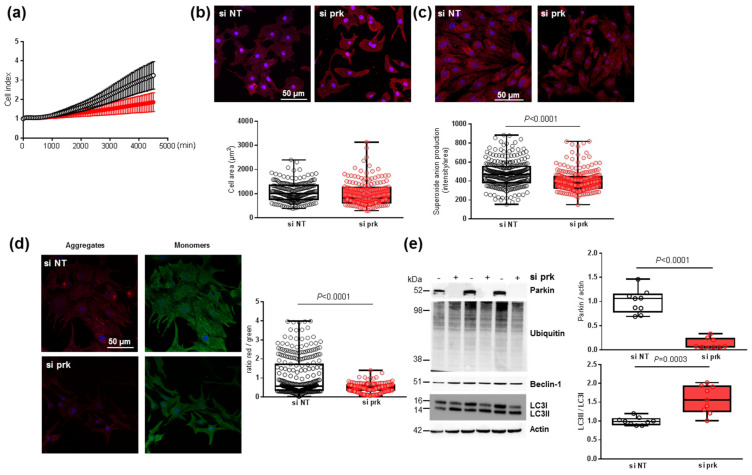
Effect of parkin depletion on hypertrophy, oxidative stress, mitochondrial biogenesis and mitophagy in neonatal rat cardiomyocytes (NCMs). (**a**) Cell index quantification by RTCA analysis in NCMs transfected with the siRNA specifically targeting rat parkin mRNA (si prk, red line) or the non-targeted siRNA (si NT, black line) as control. Cell index was recorded every 15 min (*n* = 2 independent isolations, in quadruplicate). (**b**) Hypertrophy was quantified by immunofluorescence of alpha-actinin (red) and nuclei (blue) and quantification of cell area (µm^2^) (from 3 independent experiments and at least 178 cells). (**c**) Mitochondrial superoxide anion was quantified by fluorescence quantification of mitoSOX (red) (from 3 independent experiments and at least 249 cells (**d**) Mitochondrial membrane potential was quantified by fluorescence quantification of JC-1 dye (aggregates (red) and monomer (green)) (from 3 independent experiments and at least 257 cells). (**e**) Mitophagy was quantified in NCMs transfected with the si prk (+) or si NT (−) as control. by Western blot of Parkin, ubiquitinated proteins, Beclin-1 and LC3II/LC3I ratio. Data were normalized to HPRT for RNA and actin for protein. Only significant *p* values are indicated from at least 3 independent experiments. Images were selected to represent the mean values of each condition.

**Table 1 antioxidants-11-00723-t001:** Quantification of the effect of SIRT3 inhibition on sirtuins and SOD2 activation in control and hypertrophied neonatal rat cardiomyocytes (NCMs) by Western blot.

	PBS			Iso		
	NT siRNA	SIRT3 siRNA	*p* Value	NT siRNA	SIRT3 siRNA	*p* Value
SIRT3	0.97 (0.7–1.3)	0.46 (0.3–0.6)	0.0023	0.96 (0.9–1.1)	0.31 (0.3–0.4)	0.0006
SIRT1	1.03 (0.6–1.3)	4.46 (0.9–2.3)	0.180	1.07 (0.8–1.2)	0.98 0.9–1.4)	0.937
SOD2	1.12 (0.7–1.3)	1.08 (0.6–1.6)	0.699	1.02 (0.8–1.2)	0.74 (0.5–0.8)	0.0281
SOD2acK68/SOD2	0.92 (0.85–1)	0.94 (0.8–1.3)	0.970	1.00 (0.8–1.2)	1.03 (0.7–1.5)	0.937
Bcl-2	1.00 (0.6–1.4)	0.62 (0.6–0.7)	0.200	1.00 (0.9–1.1)	0.82 (0.5–1.1)	0.400

Quantification of Western blot. Data are expressed as median with interquartile ranges.

**Table 2 antioxidants-11-00723-t002:** Quantification of the effect of SIRT3 overexpression on sirtuins and SOD2 activation in control and hypertrophied neonatal rat cardiomyocytes (NCMs) by Western blot.

	PBS			Iso		
	Ctl	SIRT3	*p* Value	Ctl	SIRT3	*p* Value
SIRT3	1.00 (0.9–1.1)	1.99 (1.1–3.2)	0.022	0.92 (0.7–1.2)	1.77 (1.7–3.2)	<0.0001
SIRT1	1.00 (0.9–1.1)	1.52 (1.1–2.6)	0.0207	1.00 (0.9–1.1)	2.21 (1.3–3.2)	0.0173
SOD2	1.00 (0.9–1.1)	1.13 (0.9–1.5)	0.461	1.00 (0.8–1.2)	0.72 (0.5–1)	0.102
SOD2acK68/SOD2	1.00 (0.9–1.1)	0.69 (0.4–0.9)	0.0195	1.00 (0.9–1.1)	0.79 (0.3–0.9)	0.0043
Bcl-2	1.00 (0.8–1.2)	2.25 (1.4–2.5)	0.057	1.00 (0.9–1.1)	1.11 (0.8–1.4)	0.886

Quantification of Western blot. Data are expressed as median with interquartile ranges.

**Table 3 antioxidants-11-00723-t003:** Quantification of the mitochondrial biogenesis and autophagy/mitophagy in hypertrophied neonatal rat cardiomyocytes (NCMs) with or without MitoQ pre-treatment.

	PBS	Iso	*p* Value *	MitoQ	*p* Value *	MitoQ + Iso	*p* Value ^#^
PGC1α	1.01 (1–1.1)	0.40 (0.3–0.6)	<0.0001	0.37 (0.3–0.6)	0.008	0.53 (0.4–1.1)	0.214
NRF2	1.02 (0.9–1.1)	0.66 (0.6–0.8)	0.0005	2.10 (1.5–2.8)	0.003	2.32 (1.9–2.9)	0.008
Mfn2	0.99 (0.9–1.1)	0.65 (0.6–0.8)	<0.0001	1.20 (1.1–1.4)	0.095	1.40 (1.2–1.6)	0.095
Fis1	0.95 (0.9–1.2)	0.64 (0.6–0.7)	0.0002	1.04 (0.9–1.1)	0.683	1.14 (0.9–1.3)	0.421
Aconitase 2	1.00 (0.9–1.3)	0.74 (0.7–0.9)	0.049	0.11 (0.1–0.4)	0.003	0.11 (0.1–0.2)	0.006
Parkin	1.00 (0.8–1.1)	0.67 (0.5–0.9)	0.014	0.17 (0.1–0.3)	<0.0001	0.19 (0.1–0.3)	<0.0001
LC3II/I	1.00 (1–1.1)	0.88 (0.7–1.1)	0.026	1.11 (0.8–1.6)	0.777	1.3 (1.1–2.3)	<0.0001
Ubiquitin	1.00 (0.8–1.2)	0.89 (0.8–1.1)	0.306	0.78 (0.6–1.1)	0.076	0.8 (0.4–1.1)	0.249
Beclin–1	1.00 (0.8–1.2)	0.82 (0.7–1.2)	0.530	0.65 (0.4–0.9)	0.0002	0.34 (0.3–0.8)	0.009

Mitochondrial biogenesis was quantified in untreated (PBS) or Iso-treated NCMs with or without MitoQ-pre-treatment by RT-qPCR of peroxisome proliferator-activated receptor gamma coactivator 1-alpha (PGC1α), Nuclear Respiratory Factor (NRF) 2, mitofusin (Mfn) 2 and Fis1. Autophagy markers were quantified in the same samples by Western blot of aconitase 2, Parkin, LC3II/I, ubiquitin and beclin-1. Data were normalized to HPRT for RNA and actin for protein and expressed as median with interquartile ranges. * compared to PBS, ^#^ compared to Iso.

## Data Availability

The data presented in this study are available in “Full unedited gel for figures” Appendix A.

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
