# Peer review of "Mitochondrial-Targeted Therapies Require Mitophagy to Prevent Oxidative Stress Induced by SOD2 Inactivation in Hypertrophied Cardiomyocytes"

_antioxidants, 2022, doi:10.3390/antiox11040723_

Round 1

Reviewer 1 Report

The article undoubtedly deserves to be published in the journal. The authors have carried out extensive research work using modern methods. The results show an ambiguous effect of mitochondrial anti-oxidant (MitoQ).

Specific commentary file attached.

Author Response

The article undoubtedly deserves to be published in the journal. The authors have carried out extensive research work using modern methods. The results show an ambiguous effect of mitochondrial anti-oxidant (MitoQ).

We thank you sincerely for your time to read our manuscript and for your following comments. We appreciate the positive comments, and we revised our manuscript in accordance to your request.

Comments:

  1. Line 211 After 6 times inversion, the tubes were centrifuged for 10 min at 500 g. What exactly is the method of inversion?

We have specified in the method (line 169): “After gently turning the tube 6 times from bottom to top”.

  1. Poor resolution of the drawings, you can't see the captions, for example:

We apologized for the poor resolution of the drawings. We have now improved the quality and resolution of all graphs in the revised manuscript.

  1. 4C, 5G-it is necessary to add clearer images (higher magnification) of mitochondrial images, for example, add them to Supplementary Materials. I would like to see «significant decrease of the mitochondria’s number but these are altered (loss of electron-dense matrix) and larger (significant increased length, width and area) (Figure 4c)», line 506-508, 570-572.

We apologized for the poor resolution of mitochondrial images. We have now increased the size of images, particularly for higher magnification, in the revised manuscript. We also added arrow to help for the interpretation.

  1. There are a lot of pictures and you can replace some of them with tables, it will relieve a huge number of visual pictures and allow to better understand the obtained data.

As suggested by the reviewer, we simplified some figures by replacing them with tables. We have now 3 tables in the manuscript and 3 supplemental tables as following:

- Table 1: Quantification of the effect of SIRT3 inhibition on sirtuins and SOD2 activation in control and hypertrophied neonatal rat cardiomyocytes (NCMs) by western blot.

- Table 2: Quantification of the effect of SIRT3 overexpression on sirtuins and SOD2 activation in control and hypertrophied neonatal rat cardiomyocytes (NCMs) by western blot.

- Table 3: Quantification of the mitochondrial biogenesis and autophagy/mitophagy in hypertro-phied neonatal rat cardiomyocytes (NCMs) with or without MitoQ pre-treatment.

- Table S1: Echocardiographic, hemodynamic and morphometric parameters in sham and HF-rats.

- Table S2: Expression of anti-oxidant and sirtuins enzymes in hypertrophied neonatal rat cardiomyocytes (NCMs).

- Table S3: Correlation between SOD2 and sirtuins in the mitochondria by PLA.

  1. You must have tested the purity of the isolated cytoplasmic and nuclear fractions. This is necessary to add to the results because even the manufacturers of NE-PER™ Nuclear and Cytoplasmic Extraction Reagents (78835, Thermo Fisher Scientific) state that the typical cross-contamination between cytosolic and nuclear fractions is 10%.

We detected a very low quantity of SIRT1, SIRT3 and GAPDH in the nuclear fraction compared to the cytosolic fraction, which could be due to cross-contamination as stated by the manufacturer. Indeed, the purity of our cytoplasmic and nuclear fractions was validated by the detection of lamin B1 only in the nuclei fraction. 

Reviewer 2 Report

In this manuscript the authors present a huge amount of data and experiments but, unfortunately, the take at home message is not so clear. Two different regulatory circuits seem to be juxtaposed, which influence the organicity and understanding of the results. On the one hand there is the regulation of superoxide production by the Sirt3/Sod2 acetylation circuit, and, on the other hand the effect of myth-Q on different processes of mitochondrial quality control is considered. Maybe focusing on one of the two pathways may help the comprehension of the text. Furthermore, the results of the different models appear to be partly contradictory and do not always support the author's conclusions.

Specific comments

-The presentation of the data is sometimes confusing. For example, fig 1e shows a reduction of Sod2 and its acetylated counterpart as well as of their ratio, which seems to be the exact opposite of what is illustrated in fig 1d. Similarly at lines 405-407 the authors describe a reduction by SOD2acK68; however some lines above, the authors report an increase in the Sod2/Sod2acK68 ratio (line 384) in the presence of unchanged levels of Sod2 (line 382), which supports the idea of an increase in SOD2acK68.

-Either Sirt3 inhibition and overexpression lead to a decrease of the area of cardiomyocytes in culture, How do the authors explain these results?

-In the in vivo model of HF, Sod2 acetylation decreases while the opposite seems to be true for the in vitro model of hypertrophy. How do the authors reconcile this discrepancy?

-Given that no differences are reported in mitochondrial respiration and ATP production between control and iso-treated cardiomyocyte which is, according to the authors, the physiological impact of the alterations in biogenesis and mitophagy observed after induction of hypertrophy?

-If Sirt3 induces Sod2 deacetylation a reduced expression of Sirt3 should be associated to increased level of acetylated Sod2, however fig 5e and lines 543-544 seem to report the opposite. Please clarify.

-Ethical approval of the study is dated january 2012,it seems quite old,  is it correct?

-Method section. How were nrc separated by ncf? How many hours were the cells exposed to the trasfection reaction? Mitochondria fraction is generally obtained by sedimentation of the organelles at 10.000-20.000 g. The authors indicate a centrifugation speed of 3000g which seems insufficient to obtain mitochondria precipitation, is it correct? Please clarify.

Author Response

In this manuscript the authors present a huge amount of data and experiments but, unfortunately, the take at home message is not so clear. Two different regulatory circuits seem to be juxtaposed, which influence the organicity and understanding of the results. On the one hand there is the regulation of superoxide production by the Sirt3/Sod2 acetylation circuit, and, on the other hand the effect of myth-Q on different processes of mitochondrial quality control is considered. Maybe focusing on one of the two pathways may help the comprehension of the text. Furthermore, the results of the different models appear to be partly contradictory and do not always support the author's conclusions.

We thank you sincerely for your time to read our manuscript and for your following remarks. We appreciate the comments, and we revised our manuscript in accordance to your request. As also suggested by the first reviewer, we simplified some figures by replacing them with tables. We have now 3 tables in the manuscript and 3 supplemental tables as following:

- Table 1: Quantification of the effect of SIRT3 inhibition on sirtuins and SOD2 activation in control and hypertrophied neonatal rat cardiomyocytes (NCMs) by western blot.

- Table 2: Quantification of the effect of SIRT3 overexpression on sirtuins and SOD2 activation in control and hypertrophied neonatal rat cardiomyocytes (NCMs) by western blot.

- Table 3: Quantification of the mitochondrial biogenesis and autophagy/mitophagy in hypertro-phied neonatal rat cardiomyocytes (NCMs) with or without MitoQ pre-treatment.

- Table S1: Echocardiographic, hemodynamic and morphometric parameters in sham and HF-rats.

- Table S2: Expression of anti-oxidant and sirtuins enzymes in hypertrophied neonatal rat cardiomyocytes (NCMs).

- Table S3: Correlation between SOD2 and sirtuins in the mitochondria by PLA.

Specific comments

-The presentation of the data is sometimes confusing. For example, fig 1e shows a reduction of Sod2 and its acetylated counterpart as well as of their ratio, which seems to be the exact opposite of what is illustrated in fig 1d. Similarly at lines 405-407 the authors describe a reduction by SOD2acK68; however some lines above, the authors report an increase in the Sod2/Sod2acK68 ratio (line 384) in the presence of unchanged levels of Sod2 (line 382), which supports the idea of an increase in SOD2acK68.

We apologized for the confusing presentation of the data. As illustrated in fig 1d, the Sod2/Sod2acK68 ratio significantly increases in vitro in the cardiomyocytes (NCM) following 24h of Iso treatment. Indeed, the sub-cellular fractionation was performed to characterize the sub-cellular localization which is qualitative and not to quantify the expression of the proteins detected. Fig 1e indicates that 1) SOD2 and its acetylated form are located in the mitochondria with SIRT1 and SIRT3 and 2) that Iso-treatment does not modify the localization. Regarding the lines 405-407, we apologized for the mistake. Our conclusion is that the decrease of the deacetylase SIRT3 in the mitochondria leads to an increase of the acetylated and inactivated form of SOD2. We modified the sentence in the revised manuscript: “These results indicate that a decrease of SIRT3 in mitochondria might be involved in the increase of SOD2acK68 observed in Iso-treated NCMs.” (lines 362-364)

-Either Sirt3 inhibition and overexpression lead to a decrease of the area of cardiomyocytes in culture, How do the authors explain these results?

We thank the reviewer for this very interesting comment. Our hypothesis is that SIRT3 deficiency might be associated with apoptosis, notably characterized by a loss of cell volume, whereas overexpression of SIRT3 could increase cell viability. We add results for Bcl-2 quantification, an anti-apoptotic protein, in the revised manuscript. Interestingly, expression of Bcl-2 tends to increase in cardiomyocytes overexpressing SIRT3, suggesting that SIRT3 could protect cell from death, notably induced by oxidative stress. This hypothesis is in concordance with results obtained in H9c2 cardiomyoblasts, in which SIRT3 inhibition is associated with an increase of acetylated (inactive) SOD2, mitochondrial ROS production and cell death (Zhao et al, Oxid Med Cell Longev., 2018, PMID: 29849892, reference 43).

We detailed this hypothesis in the revised manuscript:” Our results suggest that SIRT3 increases SOD2 activity, notably by decreases it acetylation on K68, to protect NCMs from Iso-induced hypertrophy and mitochondrial oxidative stress. These data are in concordance with previous result showing that SIRT3 inhibition in H9c2 cardiomyoblasts increased acetylated SOD2 and increased mitochondrial ROS production and cell death [43]. In NCMs, we observed that SIRT3 inhibition increased mitochondrial ROS production but decreased cell area, that might be explained by increased cell death. Indeed, SIRT3 overexpression decreased mitochondrial ROS production as well as cell area and tended to increase the anti-apoptotic Bcl2 protein. Moreover, SIRT3 inhibition also seems to blunt mitophagy under hypoxia in cardiomyoblasts [43]. Overexpression of SIRT3 increased SOD2 activity, attenuated ROS production and improved mitochondrial bioenergetics in doxorubicin-treated H9c2 cardiomyoblasts, suggesting that SIRT3 activation could be a potential therapy for some cardiac dysfunction [36].” (lines 848-866)

-In the in vivo model of HF, Sod2 acetylation decreases while the opposite seems to be true for the in vitro model of hypertrophy. How do the authors reconcile this discrepancy?

If the common phenotype associated with HF is the development of cardiac hypertrophy, several mechanisms implicating the other cardiac cell types are also involved, such as cardiac fibrosis or inflammation. In our in vitro experiment, we only focussed on cardiomyocytes hypertrophy, because mitochondria occupy at least 30% of cardiomyocyte volume. In this model, we observed that hypertrophy is associated with mitochondrial superoxide anion production, in part due to SOD2 inactivation. We previously showed in another model of hypertrophied cardiomyoblasts H9c2 cells, an increase of mitochondrial superoxide anion after 24h of Iso whereas after 48h the mitochondrial superoxide anion decreased and SOD2 expression increased, suggesting that SOD2 could be activated after long-term hypertrophy (Dubois-Deruy et al, Sci Rep, 2017, PMID: 29116107). In the in vivo model, the activation of SOD2, by decreasing acetylation, after 2 months of coronary ligation, is coherent with our hypothesis regarding a late activation of SOD2. However, we could not discriminate the cell types responsible of this activation in total left ventricle extract and could not eliminate a potential role of fibroblasts. We detailed this hypothesis in our discussion (lines 832 – 843): “In the in vivo model, we observed an activation of SOD2, by the decrease of acetylation, after 2 months of coronary ligation, which could be due to a late activation of SOD2. Indeed, we previously described an increase of mitochondrial superoxide anion after 24h of Iso in cardiomyoblasts H9c2 whereas the mitochondrial superoxide anion decreased and SOD2 expression increased after 48h, suggesting that SOD2 could be activated after long-term hypertrophy [8]. However, we could not discriminate the cell types responsible of SOD2 expression and activation in total left ventricle extract and could not eliminate a potential role of fibroblasts in in vivo results. Here, we demonstrate that K68 acetylation site is the one involved in cardiac SOD2 inactivation leading to mitochondrial superoxide anion production and hypertrophy in NCMs and that SOD2 activation by deacetylation could be involved in HF post-MI.”

-Given that no differences are reported in mitochondrial respiration and ATP production between control and iso-treated cardiomyocyte which is, according to the authors, the physiological impact of the alterations in biogenesis and mitophagy observed after induction of hypertrophy?

We agree with your remark. We observed that mitochondrial respiration is not modulated after 24h of Iso-treatment but decreased after 48h (P=0.098) (see Figure S3a) , probably as a consequence of alterations in biogenesis and mitophagy observed after induction of hypertrophy.

-If Sirt3 induces Sod2 deacetylation a reduced expression of Sirt3 should be associated to increased level of acetylated Sod2, however fig 5e and lines 543-544 seem to report the opposite. Please clarify.

We agree with your remark. Our hypothesis is that MitoQ altered mitochondrial structure and content, reflected by the decreased expression of several mitochondrial proteins such as SOD2acK68, SIRT3 and aconitase 2, the total inhibition of mitochondrial respiration quantified by oxygraphy and the alteration of mitochondria observed in electronic microscopy. It seems that MitoQ altered all the mitochondrial content, including SOD2 and SIRT3. In this case, the key point is that we observed more an alteration of mitochondrial proteins expression rather than an interaction between SOD2 and SIRT3.

-Ethical approval of the study is dated january 2012,it seems quite old, is it correct?

We confirmed that our approval was granted by the institutional ethics review board (CEEA Nord Pas-de-Calais N°242011, January 2012) for the in vivo experiments.

-Method section. How were nrc separated by ncf? How many hours were the cells exposed to the trasfection reaction? Mitochondria fraction is generally obtained by sedimentation of the organelles at 10.000-20.000 g. The authors indicate a centrifugation speed of 3000g which seems insufficient to obtain mitochondria precipitation, is it correct? Please clarify.

We apologized for having not being clear enough. For primary culture, NCMs and NCFs were separated by discontinuous Percoll gradient. At the end of centrifugation, NCF were on the top of the Percoll and NCMs between the 2 phases. We specified in the manuscript line 143: “NCMs and NCFs were separated from cell suspension by centrifugation 30 min at 3,000 g in a discontinuous Percoll gradient (bottom 58.5%, top 40.5%, Sigma-Aldrich).” For transfection, cells are transfected for 72h as indicated in line 172. Based on previous work (Fazal et al, Circ Res, 2017, PMID: 28096195), we centrifuged at 3000g but we confirmed the purity of our fractionation by detection of only ATP synthase α and SOD2 in the mitochondrial fraction, 2 proteins well described as being strictly mitochondrial. These 2 proteins were not detected in cytoplasm fraction.

Round 2

Reviewer 1 Report

Thanks to the authors for the work done

Reviewer 2 Report

The manuscript have been greatly improved and the authors have satisfactorily replied to all my comments. I have no more questions.